

# Impact of space charge on neutralization efficiency of highly charged aerosols: analytical and numerical insights using bipolar ions

Kunal Ghosh[a, b], Rukhsar Parveen[c], and Yelia Shankaranarayana Mayya[c]

[a]Department of Civil Engineering, Indian Institute of Technology Kanpur, Kanpur 208016, India
[b]Current affiliation: School of Earth and Environment,University of Leeds, LS2 9JT, United Kingdom
[c]Department of Chemical Engineering, Indian Institute of Technology Bombay, Powai 400 076, India

**Correspondence:** Dr. Kunal Ghosh (k.ghosh@leeds.ac.uk, ghosh.kunal198@gmail.com)

**Abstract.**

We investigate the influence of space charge on aerosol discharging behavior, emphasizing its importance in understanding and optimizing aerosol neutralization processes. Our study introduces the concept of space charge-assisted ionization and the distribution of charged aerosols. Specifically, we consider the case of highly charged aerosols introduced into a cylindrical
chamber containing a bipolar ion source. In this scenario, space-charge-induced motion is explicitly included in solving the dynamical equation using a computational fluid dynamics (CFD) approach. This method allows us to assess the efficacy of corona-based neutralizers for charged aerosols. Our exploration focuses on the effect of spatial ion heterogeneity induced by space charge. Our results demonstrate that the efficiency of charge neutralizers in mitigating high concentrations of charged aerosols is influenced by various factors, including aerosol charge concentration, magnitude of the charge, and design parame-
ters such as flow rate, ion production rate, and neutralizer geometry. We observe that particles at the periphery of the chamber experience a significantly slower neutralization compared to those flowing along the axis. Consequently, the aerosol system as a whole exits the chamber with a residual charge. This incomplete neutralization, caused by space charge effects, alters the particle charge distribution, potentially affecting size distribution measurements using mobility sizing instruments. These findings underscore the need to bridge theoretical concepts with practical applications in aerosol science and technology, with
broad implications for environmental monitoring and industrial processes. Furthermore, our coupled model offers potential for investigating processes involving charged aerosols in the atmosphere, where space charge effects are also significant.

## 1 Introduction

The most reliable and precise method for online sizing of sub-micrometer aerosols lies in electrical mobility analysis. Differential Mobility Analyzers (DMA) furnish the electrical mobility distribution of aerosols (Knutson and Whitby, 1975), enabling
the inference of particle size distribution when the particles charge distribution is known (Adachi et al., 1985; Reischl, 1991). Aerosols become charged within a device where ions are generated through various available techniques, primarily falling into two categories: electric discharge (direct or alternating current, dielectric barrier discharge) (Liu and Pui, 1975; Cheng





et al., 1997; Mathon et al., 2017) and irradiation ($\alpha$, $\beta$, soft X-ray, UV) (Adachi et al., 1985, 1989; Lee et al., 2005; Li et al., 2011). Irrespective of the ion generation method, particles emitted from the ionizer can carry either unipolar (of one polarity) or bipolar (of both polarities) charge. In a bipolar-ion environment, two concurrent processes occur: the charging of neutral particles and the neutralization of charged ones. Neutralization of charged particles by ions of opposite polarity typically supersedes the charging of neutral particles, resulting in relatively low charging efficiency, particularly below 10 nm, typically less than 5 % (Adachi et al., 1985; Alonso et al., 1997). Neutralization of highly charged aerosols such as the droplets produced by electrosprays are particularly challenging (Singh et al., 2016). Neutralization of the charge on aerosol particles refers to the process in which their initial charge is brought to zero under the action of bipolar charging (Alonso et al., 1997; Kimoto et al., 2009; Carsí and Alonso, 2020). This is an ideal situation obtained under strict symmetric charging conditions, in which case the aerosols attain a stationary charge distribution with mean zero regardless of original distribution. In practice, a slight difference in the ionic mobilities would induce a small, non-zero mean charge. However, this may be ignored if the ion source is strong. The bipolar charging and neutralization are generally achieved by radioactive sources (Adachi et al., 1985, 1989; Lee et al., 2005; Li et al., 2011) or, in recent times, by AC corona chargers (Liu and Pui, 1975; Cheng et al., 1997; Mathon et al., 2017). The study of this process has been of considerable theoretical interest in view of its fundamental nature and importance in several practical applications. Applications include the elimination of unwanted initial charges on test aerosols Ghosh et al. (2017), environmental aerosol charges in the context of atmospheric electricity (Brattich et al., 2019; Miller et al., 2024), and, most importantly, the imparting of a specified distribution in electrostatic size analyzers (Knutson and Whitby, 1975; Adachi et al., 1985; Reischl, 1991). The development of neutralizers for optimal performance requires understanding of various effects responsible for the neutralization process.

An important parameter in neutralization process is the mean relaxation time required to attain the stationary distribution from an unknown initial charge distribution. The estimate of this time is provided by the time taken to reduce the initial mean charge of the particles to a prescribed fraction, say, 1/e of the original value. Much of the classical neutralization theories are based on dilute aerosol models, or what may be termed as single particle models (Fuchs, 1963; Hoppel and Frick, 1986). In this, we assume that the aerosol particle is immersed in a specified ionic environment and the rate of loss of charge on the particle is governed essentially by the $Nt$ product. On the basis of this, Liu et al. (1986) estimated the neutralization times essentially in continuum limit. Later, Mayya and Sapra (1996) obtained a neutralization coefficient in the entire particle size range using the combination coefficients of Hoppel and Frick (1986). More recently, Alonso (2018) proposed an exact formula for the asymptotic neutralization rate for particles in the free molecular size range. All these theories essentially demonstrated that smaller particles require larger residence times for neutralization.

The diluted aerosol limit is essentially an ideal limit, and in practice two types of particle concentration effects appear. The first one, the ion-depletion effect in which the concentration of ions gets lowered as a result of attachment to aerosols and hence the neutralization times, would necessarily depend upon the particle concentrations, in addition to size. This effect was formally examined in detail by Hoppel and Frick (1986, 1990) by including the attachment rates as a part of the ion balance equations and forms the basis of the design of bipolar charging devices. The most significant result is that at high aerosol concentrations (Z), the ion densities would decrease as 1/Z, and hence the relaxation rates would decrease proportionately.





The second effect pertains to the effect of space charge. This matter was discussed in Hoppel and Frick (1986, 1990) under an asymmetric charging situation as a part of a supplementary equation for obtaining the mean charge under steady state. However, if the initial aerosol carried a net charge, then the space charge would be mainly due to this charge itself than due to ion asymmetry. The collective space charge would strongly alter the ion profiles, thus affecting the neutralization times. This work attempts a model study of the possible implications of the space charge effects in governing aerosol discharging behavior.

The fundamental motivation for this study stems from the fact that, there has been increasing interest in the generation of concentrated charged aerosol dispersion by various techniques. It is also known that much smaller particles generated by plasma processing (Kim, 2005), hot wire generators (Ghosh et al., 2021) and electrostatic sprays (Tang and Gomez, 1994) carry very high charges. It is generally recognized that concentrated aerosols made of particles carrying high charges cannot be easily neutralized by conventional radioactive sources and this has in turn given a large impetus to coronal based bipolar chargers (Ibarra et al., 2020). An understanding of the basic discharging mechanisms of such systems would be helpful in understanding the performance of chargers intended for high concentration aerosols. We present a comprehensive analysis of the effects of space charge on aerosol discharging behavior. Through theoretical modeling, we explore the spatial heterogeneity of ions and its impact on neutralization rates. Our findings not only enhance our understanding of aerosol dynamics but also pave the way for the design of novel neutralization devices capable of addressing the challenges posed by high-concentration of charged aerosols.

## 2 Theoretical and numerical analysis

We examine the charging and neutralization of monodisperse aerosol particles in a circular tube of radius R and length L, in which ion pairs are continuously generated at a volumetric rate S. For simplicity, the generation rate S is considered uniform all along and across the tube, although this situation does not occur in practice (Alonso and Alguacil, 2003). We considered the ions are generated steadily by a radiation source and pass highly charged aerosols at t=0. At the outlet of the tube, we examine whether the average charge of all aerosols is reduced to the pre-specified low level, as expected by the design of the instrument.

### 2.1 Ion and charge aerosol dynamics:

Let us first assume that particles are of much larger size compared to ions. That means, in the time scales of interest, ion motion and particle motion, either by diffusion, or by electromigration, may be neglected. We also neglect turbulence. Then the general forms of the ion dynamics equations are:

$$\frac{\partial n^+(x_i,t)}{\partial t} + u_i \bigtriangledown .n^+(x_i,t) + \mu^+ \bigtriangledown .En^+(x_i,t) - D^+ \bigtriangledown^2 n^+(x_i,t) = S - \alpha n^+(x_i,t)n^-(x_i,t) - n^+(x_i,t) \sum_{q=-\infty}^{\infty} Z(x_i,q,t)\beta_q^+ \tag{1}$$

where $n^+$ is the concentration of positive ions, and $n^-$ is the concentration of negative ions. $\beta$ is the ion aerosol attachment coefficient, $\alpha$ is the ion ion recombination coefficient and $Z$ is the charge aerosol concentration. $x_i$ represents direction for $X$





and $Y$, $u_i$ represents corresponding velocities and $\rho_{air}$ denotes the density of air. and

$$\frac{\partial n^-(x_i,t)}{\partial t} + u_i \bigtriangledown .n^-(x_i,t) - \mu^- \bigtriangledown .En^-(x_i,t) - D^- \bigtriangledown^2 n^-(x_i,t) = S - \alpha n^+(x_i,t)n^-(x_i,t) - n^-(x_i,t) \sum_{q=-\infty}^{\infty} Z(x_i,q,t)\beta_q^- \qquad (2)$$

The equation for space charge is then given by,

$$\bigtriangledown .E = \frac{e}{\epsilon_0}[n^+(x_i,t) - n^-(x_i,t) + \sum_{q=-\infty}^{\infty} Z(x_i,q,t)\bar{q}] \qquad (3)$$

considering $curl E = 0$, the charging equation for a mono-disperse aerosol is,

$$\frac{\partial Z(x_i,q,t)}{\partial t} + u_i \bigtriangledown .Z(x_i,q,t) = n^+(x_i,t)\beta_{q-1}^+ Z(x_i,q-1,t) + n^-(x_i,t)\beta_{q+1}^- Z(x_i,q+1,t) - [n^+(x_i,t)\beta_q^+ + n^-(x_i,t)\beta_q^-]Z(x_i,q,t) \qquad (4)$$

In the absence of a 'space charge' condition, it can be demonstrated that aerosols cannot be completely neutralized, even in a perfectly bipolar ion atmosphere—a phenomenon we refer to as the 'neutralization conundrum.' In their seminal papers, Hoppel and Frick (1986, 1990) formulated the neutralization problem in great detail. While these studies examined various influencing factors, the fundamental role of space charge in driving the neutralization process has not been sufficiently emphasized. As we demonstrate in the Appendix A, the presence of a space charge, induced by a non-zero mean charge on the aerosol—however small—is crucial for achieving quasi-neutrality in the system. If space charge is excluded, under the assumption that it is merely a corrective factor, the mean charge of the aerosol fails to decay to zero, even under conditions of perfectly symmetric ion mobilities. This conundrum arises because an infinitesimally small effect plays a pivotal role in facilitating aerosol neutralization.

## 2.2 Inclusion of CFD on charge aerosol dynamics:

It is known that in realistic charge neutralizers we have some input flow (Wang and Flagan, 1990), so we consider the inclusion of CFD equations (Ghosh et al., 2020) for calculating the flow velocity as,

$$\frac{\partial(\rho_{air}u_i)}{\partial t} + \frac{\partial(\rho_{air}u_iu_j)}{\partial x_i} = -\frac{\partial P}{\partial x_i}, \qquad (5)$$

where $P$ is the system pressure (same as atmospheric pressure here), $u_i$ represents corresponding velocities and $\rho_{air}$ denotes the density of air. At t=0 we consider input velocity ($u_i$) calculating from input flow rate and other input condition given in Table 1. Then the general transport equation for charge aerosol can be written as

$$\frac{\partial Z(x_i,q,t)}{\partial t} + \frac{\partial u_i Z(x_i,q,t)}{\partial x_i} + \mu \bigtriangledown .EZ(x_i,q,t) = D \bigtriangledown^2 Z(x_i,q,t) \qquad (6)$$

where $u_i$ is the flow velocity, $Z(x_i,q,t)$ the concentration number of the charge aerosol, $\mu$ its electrical mobility, $E$ the electric field, $D$ the diffusion coefficient. In typical neutralizer setups, ions are generally assumed to be well-mixed relative to aerosols, which allows us to disregard the effects of flow on ion dynamics (Ibarra et al., 2020; Wang and Flagan, 1990). As a result, we focus on the aerosol dynamics where the flow plays a more significant role.



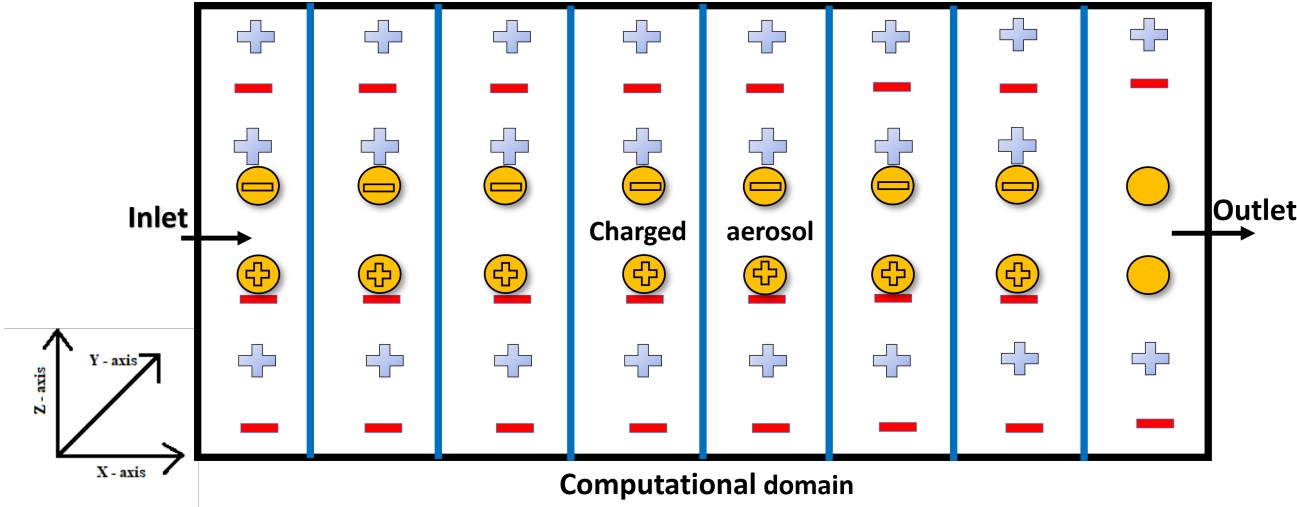

**Figure 1.** A simplified representation of commonly used charge neutralizer, the computational domain showing the space bins and the movement of ions and charged aerosols through them.

## 2.3 Modelling setup

We use a 2D geometry to minimize computational costs, a practice justified by the predominant flow being propelled by the flow rate of the neutralizer (forced convection). The simplified representation involves a 2D version of a cylindrical charge neutralizer, illustrated in Figure 1. In this scenario, charged aerosols are presumed to enter the chamber alongside flowing air, and ion production is considered to be homogeneous throughout the chamber. To streamline the calculations for 2D fluid flow, fluid properties were estimated in the X-Z plane and subsequently merged with the aerosol dynamical equation at designated

nodes (Lu et al., 1996). The chamber was discretized into small variable grids with an incremental ratio of 1, ensuring a minimum grid size of 1 mm × 1 mm at the center of the chamber. The number of grids in the X and Z directions remained constant, dependent on the size of the enclosure. Subsequently, the Navier-Stokes equations, describing fluid motion, were applied for a given set of boundary conditions. These equations, in conjunction with the turbulence, energy equation, and aerosol dynamics, were solved at each grid of the mesh. All simulations were conducted under the influence of external forced

flow, assuming different flow rates. The variation in temperature throughout the chamber was considered constant, eliminating flow related to natural convection. To obtain precise numerical solutions, meticulous meshing of the computational domain was crucial, particularly for the boundary layer, where steep gradients in flow dynamics create a highly convective zone, as shown in Figure 2. Consequently, an enhanced wall function has been incorporated to enhance accuracy near the boundary layer. The model simulations focused on the sizes of monomer aerosols, specifically 1 μm. The model input parameters are shown in Table

130 1





**Table 1.** Parameters used for model calculation

| Item | Parameter name | Parameter value |
|---|---|---|
| Neutralizer | Length (L) | 5 & 10 cm |
| | Radius (R) | 2 & 3 cm |
| | flow rate | 1, 2, 3 & 5 lpm |
| | Ion production rate | $10^4$, $10^5$, $10^6$ and $10^7$ ion$-$cm$^{-3}$/sec |
| | Aerosol concentration | $10^3$, $10^4$, $10^5$ and $10^6$ number/cm$^3$ |
| | Aerosol charge level | +10 |

## 3  Results and Discussion

### 3.1  Simulation of flow and ion profile inside cylinder

#### 3.1.1  Air flow profile

Figure 2 illustrates the model-predicted steady-state velocity profiles corresponding to four different flow rates: 1 lpm, 2 lpm, 3
lpm, and 5 lpm. All flow rates align with those of a commercially available neutralizer (Ibarra et al., 2020). The X-axis represents
the horizontal distance from the inlet to the outlet of the cylinder, while the Z-axis represents the vertical distance between the
lower and upper boundaries of the cylinder. In all cases of flow rates, the velocity profiles exhibit similar behavior, with the
maximum velocity observed along the center line and the minimum velocity at the upper and lower boundaries, influenced by
the boundary layer effect on airflow. This airflow carries charged particles throughout the cylinder, resulting in varying times
for ions to neutralize and impacting the efficiency of the neutralizer, as discussed in subsequent results.



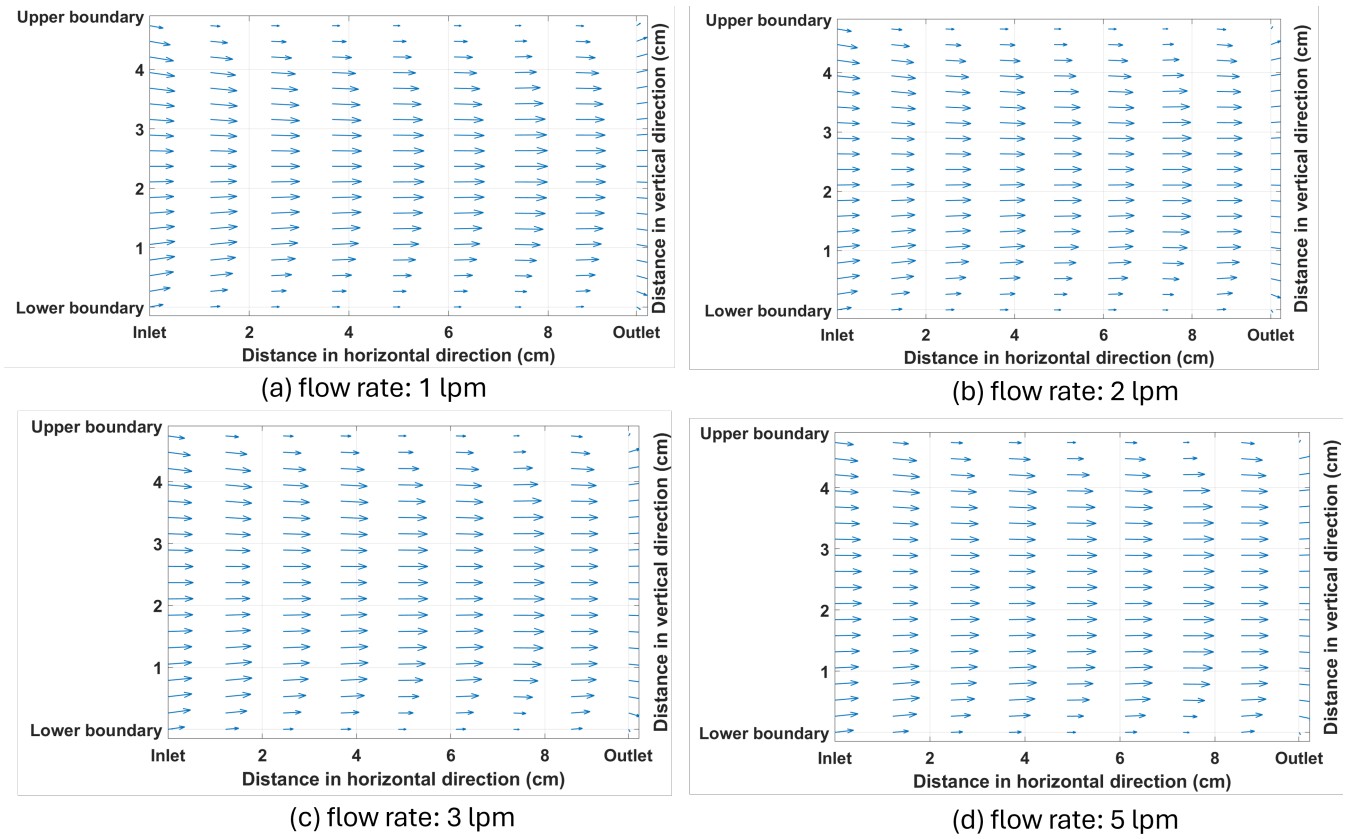

(a) flow rate: 1 lpm

(b) flow rate: 2 lpm

(c) flow rate: 3 lpm

(d) flow rate: 5 lpm

**Figure 2.** Model-predicted steady-state velocity profiles for a cylindrical geometry under four different flow rates



### 3.1.2 Ion profile

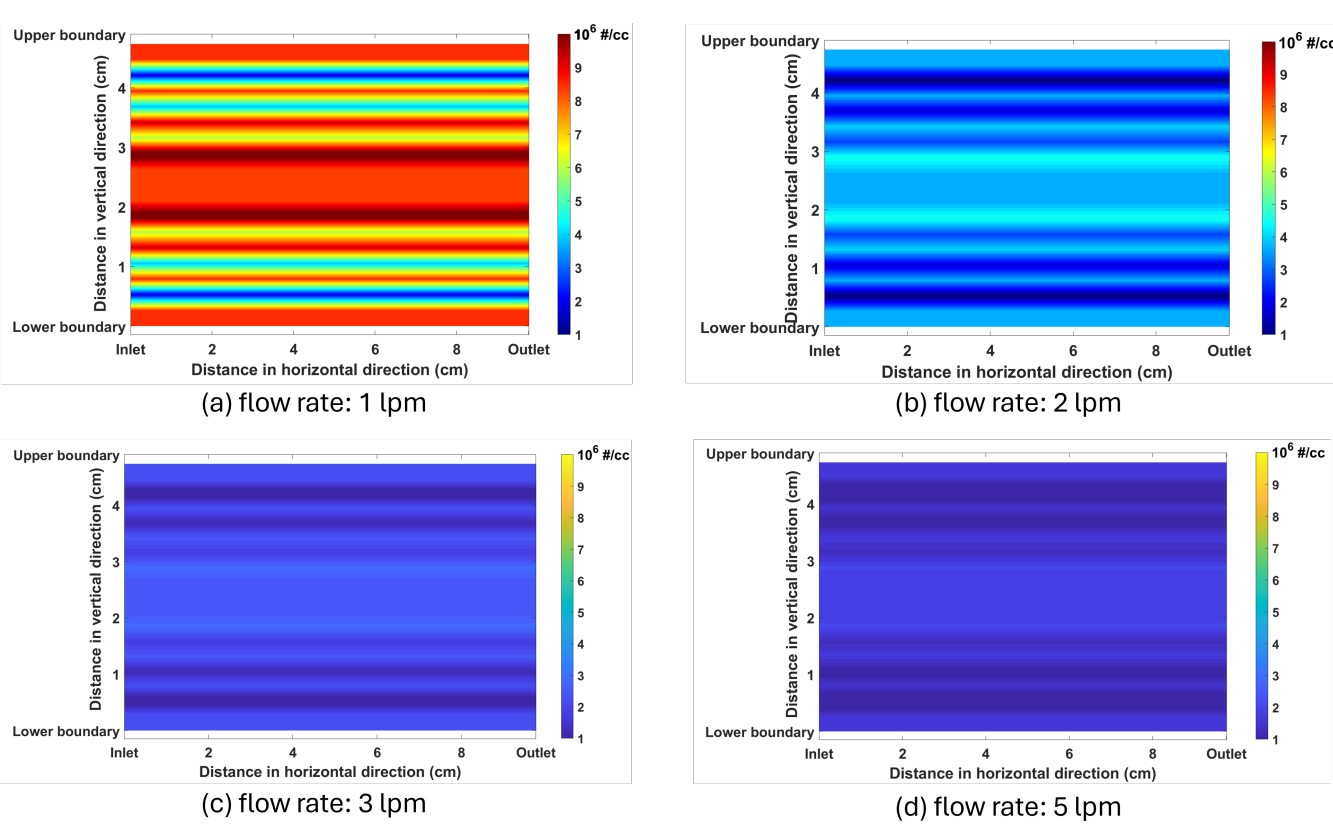

**Figure 3.** Steady-state ion profiles for four different flow rates in a two-dimensional space

Figure 3 illustrates the model-predicted steady-state positive ion profiles subsequent to the passage of positively charged aerosols (concentration = $10^4$ number/cm$^3$) through the cylinder under four distinct flow rates: 1 lpm, 2 lpm, 3 lpm, and 5 lpm. For each flow scenario, a constant ion production rate of $\approx 10^6$ number/cm$^3$ − sec is maintained throughout the cylinder. The ion concentration along the Z axis exhibits variations as a result of space-charge effects. The maximum ion concentration is observed at the center, with concentrations decreasing towards both upper and lower boundaries. This phenomenon arises from the electric field generated by the boundaries, which displaces the ion concentration, resulting in varied concentrations. The fluctuating ion concentration also impacts the neutralizer capacity of charged aerosols. Additionally, an increase in the flow rate corresponds to a lesser variation in ion concentration, attributed to the reduced residence time of charged aerosols within the chamber, thereby generating a weaker electric field and consequently less variation in the ion concentration profile.



## 3.2 Neutralizer effect of charge aerosol

In this section, we discuss the model-predicted results of neutralizer capacity in the presence of charged aerosols. To comprehend the space charge effect on neutralizer capacity, we simulated our model under four different neutralizer conditions.

### 3.2.1 Effect of flow on neutralizer capacity

The first aspect we examine is the effect of flow on the charge neutralizer capacity, while maintaining other parameters constant, such as ion production rate ($10^6$ ion/cm$^3$ − sec), aerosol concentration ($10^4$ number/cm$^3$), charge level of aerosol (+10), and geometric dimensions (length = 10 cm and diameter = 6 cm). To explore this, we consider four different flow rates: 1 lpm, 2 lpm, 3 lpm, and 5 lpm. The results of different flows on the average charge on aerosol particles are shown in Figure 4.

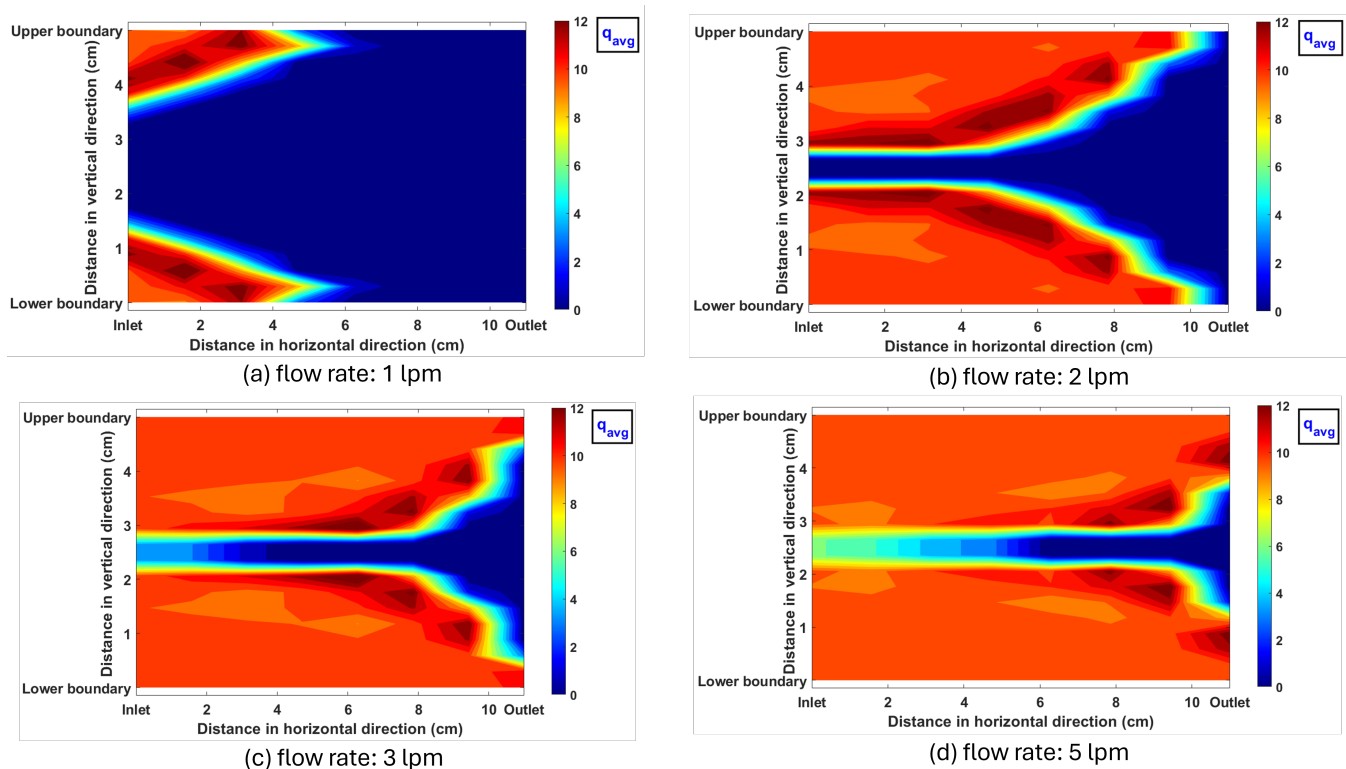

**Figure 4.** Steady state spatial average charge on particle profile in in a two-dimensional space for four different flow rate of aerosol

The models predicts that for all four flow rates considered, the charge on the particles at the center are neutralized the fastest,
while those near the boundaries are neutralized the slowest. Moreover, particles near the wall initially experience an increase in charge (counter-neutralization) before being neutralized. This phenomenon arises because of the space-charge effect, where positively charged particles generate an electric field that drives positive ions towards the periphery, leading to further unipolar



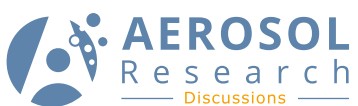

charging of particles. Over time, particles become neutralized at the center, resulting in a decrease in the space charge effect, which eventually causes the decay of peripheral particle charges.

With an increase in the flow rate, particles have to travel a greater distance before neutralization occurs, because of the reduced residence time of particles inside the neutralizer. For a flow rate of 1 lpm, particles begin to neutralize fully from the inlet at the center at around 4 cm from the inlet. For a flow rate of 2 lpm, charge particles are neutralized quite close to the outlet, around 9 cm from the inlet. For flow rates of 3 and 5 lpm, neutralization occurs closer to the outlet along the centerline. At the boundary, however, particles are not fully neutralized. For a flow rate of 3 lpm approximately 20 % of particles remain charged

at the outlet, wheres for 5 lpm flow rate, approximately more than 50 % of the particles are charged at the outlet. Therefore, for flow rates of 3 lpm and above, the neutralizer efficacy is less than expected, when the effect of space charge is considered under the given simulation conditions.

### 3.2.2   Effect of ion-production on neutralizer capacity

In this section, we examine the effect of ion production on the charge neutralizer capacity while maintaining other parameters

constant, such as the flow rate (2 lpm), aerosol concentration ($10^4$ number/cm$^3$), charge level of aerosol (+10), and geometric dimensions (length = 10 cm and diameter = 6 cm) of the neutralizer. We consider four different ion production rates: $10^4$ ion/cm$^3$ − sec, $10^5$ ion/cm$^3$ − sec, $10^6$ ion/cm$^3$ − sec, and $10^7$ ion/cm$^3$ − sec. The ion production rate is also selected according to the specifications found in commercially available neutralizers (Adachi et al., 1985; Reischl, 1991). The results of varying ion production rates on the average charge of aerosol particles are shown in Figure 5.

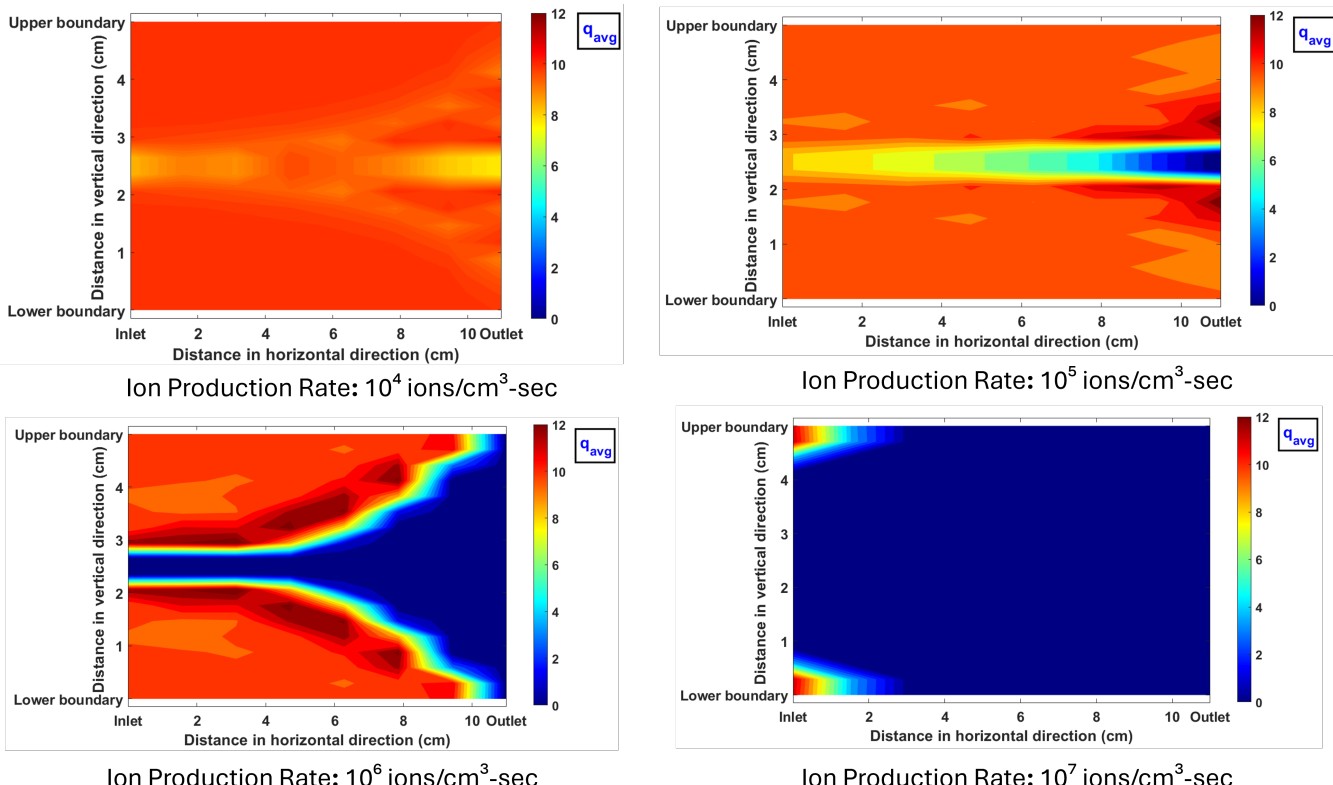

**Figure 5.** Steady state spatial average charge on particle profile in in a two-dimensional space for four different ion production rate

At the given flow rate of 2 LPM, ion production rates of $10^4$ ions/cm³ – sec and $10^5$ ions/cm³ – sec are insufficient to fully neutralize the charged particles. In the $10^5$ ions/cm³ – sec case, counter-neutralization due to space charge is also observed, leading to an increase in the average charge of aerosol particles within 1 cm above and below the centerline. However, at ion production rates of $10^6$ ions/cm³ – sec and $10^7$ ions/cm³ – sec, complete neutralization occurs before the particles reach the outlet. The highest level of neutralization is achieved at the higher ion production rates, which can be attributed to the increased availability of ions for neutralizing the charged aerosols.

### 3.2.3  Effect of aerosol concentration on neutralizer capacity

In this section, we examine the effect of varying charged aerosol concentration on the charge neutralizer capacity while maintaining other parameters constant, such as the flow rate (2 lpm), charge level of aerosol (+10), and ion production rate $10^6$ ions/cm³ – sec, and geometric dimensions (length = 10 cm and diameter = 6 cm) of the neutralizer. We considered four type of aerosol concentration based on different purposes of this type of instrument are use: a) $10^3$ number/cm³ and b) $10^4$ number/cm³ is for atmospheric aerosols (McMurry, 2000), c) $10^5$ number/cm³ and d) $10^6$ number/cm³ are selected represents for industrial

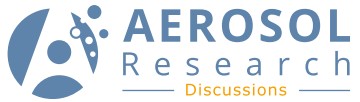

sources like car exhaustion (Wehner et al., 2009), chimneys emissions (Chen et al., 2017), hot wire generator source (Ghosh et al., 2020, 2021).

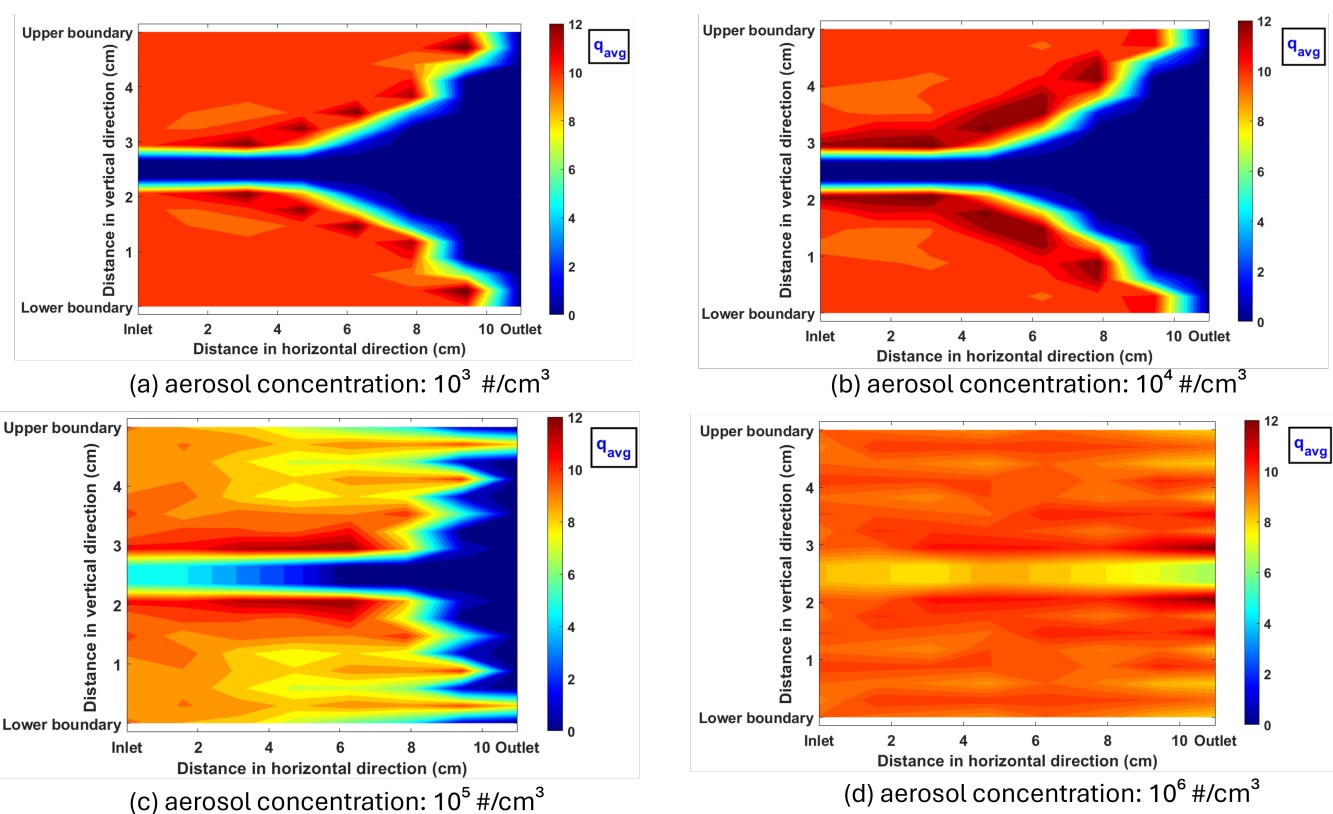

**Figure 6.** Steady state spatial average charge on particle profile in in a two-dimensional space for four different charged aerosol concentration

The Figure 6 illustrates the steady-state average charge on particles for four different aerosol concentrations: $10^3$ number/cm$^{-3}$, $10^4$ number/cm$^{-3}$, $10^5$ number/cm$^{-3}$, and $10^6$ number/cm$^{-3}$. As the charged aerosol concentration increases, fewer ions will be available for neutralization, resulting in less effective neutralization for higher aerosol concentrations. For an aerosol concentration of $10^5$ number/cm$^{-3}$, we observe a increase in irregular neutralization at the boundaries compared to the center line. This is attributed to the presence of highly charged aerosols, which increase the space charge (Eq. 3), causing more positive ions to migrate to the boundaries, thereby increasing unipolar charging. Additionally, the flow velocities at the boundaries are lower than at the center (as shown in Figure 2), allowing the aerosols more residence time for neutralization. These two counteracting effects result in fluctuations in the neutralization of charged aerosols. A similar trend is observed for an aerosol concentration of $10^6$ number/cm$^{-3}$. It is suggested that for high-charge aerosol concentrations considered in the charge neutralization process, the classical "$Nt$" product concept does not hold. It also depends on the aerosol concentration and initial charge. Therefore, for higher aerosol concentrations, the efficacy of the neutralizer under the current conditions is very less. The efficacy of the

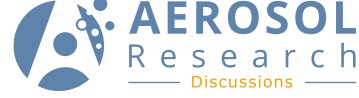



neutralizer may be improved by adjusting the other designing parameters of the neutralizer like flow rate, ion production rate
and geometry.

### 3.2.4  Effect of geometry on neutralizer capacity

In this section, we examine the effect of different type of geometry on the charge neutralizer capacity while maintaining other
parameters constant, such as the flow rate (2 lpm), aerosol concentration ($10^4$ number/cm$^3$), charge level of aerosol (+10), and

ion production rate $10^6$ ions/cm$^3$ − sec of the neutralizer. We consider four different combination of length and width of the
neutralizer consistent with commercially used (Ibarra et al., 2020).

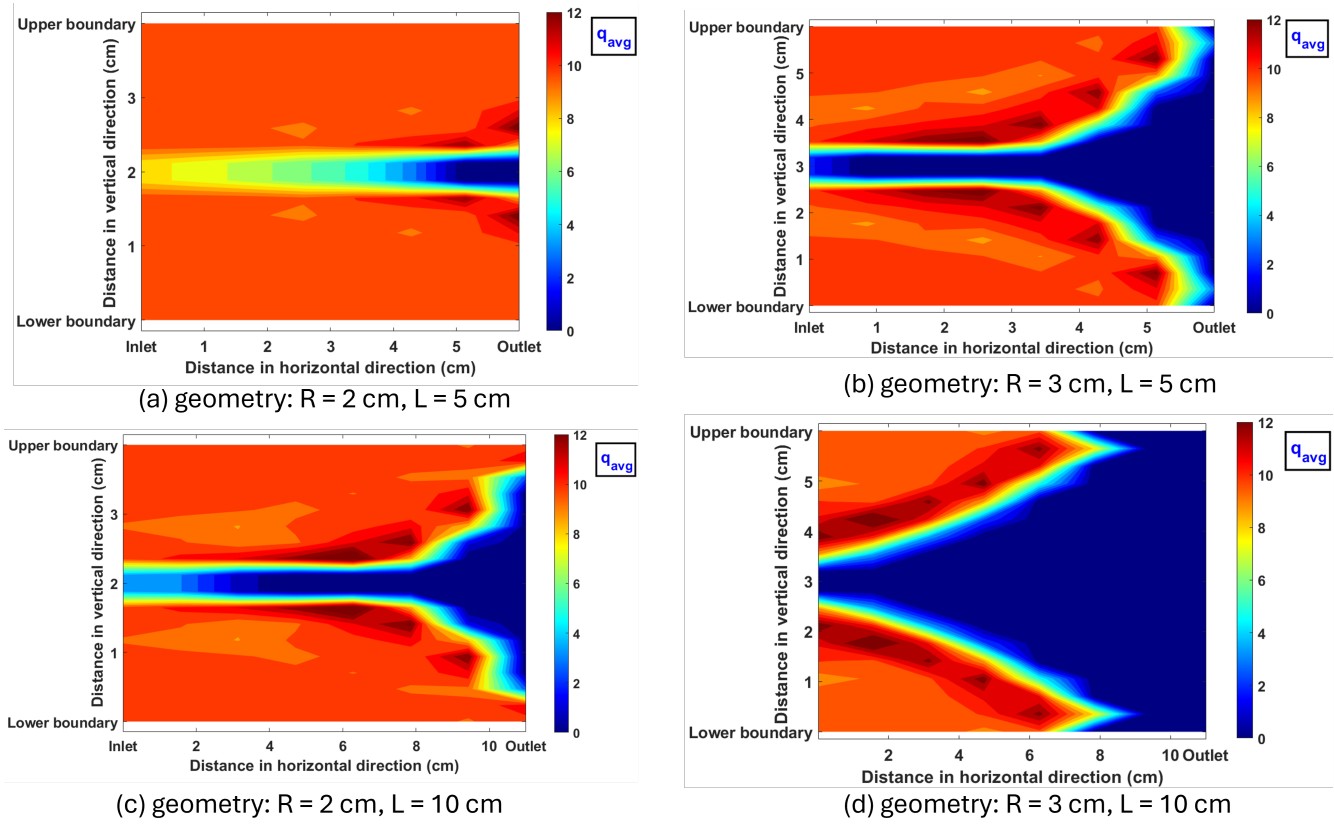

**Figure 7.** Steady state spatial average charge on particle profile in in a two-dimensional space for four different aerosol neutralizer geometry

Figure 7 shows the steady-state average charge distribution across particles across four distinct neutralizer geometries, de-
lineated by their respective radius (R) and length (L): (a) R = 2 cm, L = 5 cm; (b) R = 3 cm, L = 5 cm; (c) R = 2 cm, L = 10
cm; and (d) R = 3 cm, L = 10 cm. Notably, for configurations sharing the same length (L = 5 cm), cylindrical neutralizers with

smaller radii (R = 2 cm) exhibit heightened air velocities in contrast to those with larger radii (R =3 cm), under identical flow
rates of 2 lpm. Consequently, this differential in air velocity results in a diminished residence time for charged aerosol particles



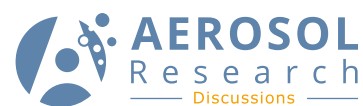

within the cylinder. Hence, neutralizers with larger radii (R = 3 cm) demonstrate superior neutralization efficiency compared to their smaller counterparts. This trend extends to neutralizer length, with elongated neutralizers affording increased duration for charge particle neutralization in comparison to shorter variants. However, it is noteworthy that for cylinders with a radius of R

= 2 cm and a length of L = 10 cm, observations at the outlet reveal incomplete neutralization of charged particles, particularly near the upper and lower boundaries, which can be attributed to the presence of space charge effects that are generally not considered during the design of this type of neutralizer.

## 4    Conclusion

We studied the effect of space charge on neutralization rate with respect to the flow rate, ion production rate, charged aerosol

concentration, magnitude of charge, and the neutralizer geometry. These are the factors which influence the working of neutralizers, specifically used for charged aerosols. We show that space charge can significantly compound the effect of these factors in lowering the efficacy of neutralizers, and in some cases, render them entirely ineffective. Increasing the flow rate decreases the residence time, resulting in less efficient neutralization. Similarly, insufficient ion production rates also lead to inefficient neutralization. Moreover, as aerosol concentration rises, neutralization becomes less effective due to fewer available ions. Due

to space charge positive and negative ions separate spatially casing differential neutralization in different regions. Infact, the charges on particles at the boundary of the channel could increase initially rather than decrease due to ion segregation. Additionally, neutralizer geometry plays a crucial role, with larger radii and lengths facilitating improved neutralization efficiency compared to smaller counterparts. However, incomplete neutralization, particularly near the upper and lower boundaries, is observed in certain configurations, indicating the presence of space charge effects. These findings highlight the importance of

considering various parameters for optimizing neutralization processes within aerosol control systems.

The broader applicability of these findings extends beyond the realm of aerosol control systems. Similar principles elucidated in this study could be relevant in various fields where charged particle dynamics are pertinent, including environmental monitoring, semiconductor manufacturing, pharmaceuticals, and atmospheric science. For instance, in environmental monitoring, understanding the behavior of charged particles could aid in designing efficient air purification systems. In semiconductor

manufacturing, where charged particles can adversely impact production processes, optimizing neutralization techniques could enhance product quality and yield. Similarly, in atmospheric science, insights into charged aerosol dynamics could improve understanding of air pollution and climate change processes. Overall, the findings from this study not only inform the design of neutralizers for aerosol control but also have broader implications for diverse applications where the management of charged particles is essential.





## Appendix A: Proof of "neutralization conundrum"

In their classic papers, Hoppel and Frick (1986, 1990) examined the issue of symmetric and asymmetric Boltzmann law from the perspective of charge neutrality condition combined with bipolar ion-aerosol attachment process. Considering a spatially homogeneous system, they argued that if the ion properties are perfectly symmetric, then the originally charged aerosol system will attain zero mean charge leading to symmetric Boltzmann distribution. The key to this result is the assumption that the combined system of ions and aerosols would attain perfect charge neutrality in space after sufficiently long time. However, as it turns out, even if the ions are completely symmetric, the mean charge of an initially charged aerosol system does not decay to zero (neutralization) by attachment mechanism alone. One has to necessarily include the space-charge effect, however small, to attain complete neutralization of an initially charged aerosol. We refer to this observation as "neutralization conundrum", i.e. a difficulty in understanding because of the conceptual need to include an effect which does not otherwise practically affect the rate of neutralization. The proof of this conundrum is as follows: Suppose an aerosol is spatially distributed uniformly at a concentration $Z$ particles per unit volume, maintained, say, inside a radioactive neutralizer having a bipolar ion production rate density, $S$. We show that even if the ions are perfectly symmetric in respect of their mobility characteristics, an initially charged aerosol will not be neutralized unless we incorporate space charge effects. In other words, the space-charge-induced drift is the guarantor of complete neutralization of charged aerosol in a symmetric bipolar ion system and not merely the symmetry of the ions. Because of the assumption of spatially homogeneity, the gradient effects of convection and diffusion may be ignored. The ion balance equations, excluding space-charge effect, but including ion asymmetry, production, recombination and attachment to aerosols, are given below in Eqs.(A1,A2). The particle charging equation is given in Eq.(A3):

$$\frac{\partial n^+}{\partial t} = S - \alpha_0 n^+ n^- - n^+ \sum_{q=-\infty}^{\infty} \beta_q^+ Z_q \tag{A1}$$

$$\frac{\partial n^-}{\partial t} = S - \alpha_0 n^+ n^- - n^- \sum_{q=-\infty}^{\infty} \beta_q^- Z_q \tag{A2}$$

$$\frac{\partial Z_q}{\partial t} = n^+ \beta_{q-1}^+ Z_{q-1} + n^- \beta_{q+1}^- Z_{q+1} - (n^+ \beta_q^+ + n^- \beta_q^-) Z_q \tag{A3}$$

The space-charge density ($\rho$) is defined as

$$\rho = n^+ - n^- + \bar{q} Z \tag{A4}$$

The mean charge ($\bar{q}$) per particle is defined using Eq.(A3) as

$$\bar{q} = \frac{1}{Z} \sum_{q=-\infty}^{\infty} q Z_q \tag{A5}$$



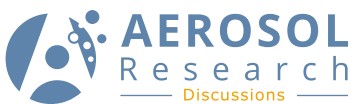

where $Z = \sum_{q=-\infty}^{\infty} Z_q$ is the total (all charges) particle concentration. From Eq.(A3) and Eq.(A5), the evolution of mean charge can be shown to satisfy the following equation:

$$\frac{d(\bar{q}Z)}{dt} = \sum_{q=-\infty}^{\infty} (n^+ \beta_q^+ - n^- \beta_q^-) Z_q \tag{A6}$$

For illustration purposes, we restrict ourselves to the case of continuum attachment coefficients:

$$\beta_q^+ = \frac{\beta_0^+ q\alpha e^{-q\alpha}}{1 - e^{-q\alpha}}, \qquad\qquad \beta_q^- = \frac{\beta_0^- q\alpha}{1 - e^{-q\alpha}}, \tag{A7}$$

where, $\alpha = \frac{r_c}{a}$, ($r_c = \frac{e^2}{4\pi\epsilon_0 k_B T}$ = Coulomb length $\approx 56$ nm at $T = 300$ K) and $\beta_0^{+-} = 4\pi\alpha D^{+-}$ are the attachment coefficients between the neutral particles and positive (+), negative (-) ions.

    An equation for the space charge density $\rho$ defined by Eq. (A4) is obtained by carefully working through Eq. (A1 to A3) & Eq.(A4 to A6). While doing so, we note that the charging terms cancel exactly and we obtain

$$\frac{d\rho}{dt} = 0 \tag{A8}$$

The solution to Eq. (A8) is $\rho(t) = constant$ at all times. If originally the particle carried a charge $q_0$, and the original ion concentrations were equal $n^+(0) = n^-(0)$, then Eq.(A8) implies that

$$\rho(t) = constant = \rho_0 = Zq_0 \tag{A9}$$

As this equation is true at all times, it should be true in steady-state ($t \to \infty$) as well. Hence in steady-state,

$$n_\infty^+ - n_\infty^- + Z\bar{q}_\infty = Zq_0 \tag{A10}$$

In steady-state, within the framework of continuum models, the solution to Eq.(A3) yields a shifted Boltzmann charge distribution having a mean charge

$$\bar{q}_\infty = \frac{1}{\alpha}[ln\frac{n_\infty^+}{n_\infty^-} + ln\frac{\mu^+}{\mu^-}], \tag{A11}$$

where $\alpha = \frac{r_c}{a}$. Then the solution for the steady-state mean charge of aerosol will be a solution of the following transcendental equation

$$\bar{q}_\infty = \frac{1}{\alpha}ln\left[(\frac{\mu^+}{\mu^-})\frac{2N_\infty + Z(q_0 - \bar{q}_\infty)}{2N_\infty - Z(q_0 - \bar{q}_\infty)}\right] \tag{A12}$$

Where $N_\infty$ is the sign independent average ion concentration

$$N_\infty = \frac{1}{2}(n_\infty^+ + n_\infty^-), \tag{A13}$$



Eq.(A12) requires to be numerically solved for $\bar{q}_\infty$. For illustrating the point that Eq.(A13) yields a nonzero solution for $\bar{q}_\infty$ even when $\mu^+ = \mu^-$, we can assume high ion density limit, i.e. $N_\infty \gg Z$. Then the terms within the log function can be linearized and one obtains,

$$\bar{q}_\infty = \frac{1}{\alpha}\left[\frac{Z}{N_\infty}(q_0 - \bar{q}_\infty)\right] + \frac{1}{\alpha}ln\gamma, \tag{A14}$$

where $\gamma = \frac{\mu^+}{\mu^-} =$ ion asymmetry factor. Eq.(A14) possesses a solution

$$\bar{q}_\infty = \frac{1}{\alpha}\left[\frac{\frac{Z}{N_\infty}q_0 + ln\gamma}{1 + \frac{Z}{\alpha N_\infty}}\right], \tag{A15}$$

From this we see that even if $\gamma = 1$, (perfectly symmetric ions), $\bar{q}_\infty$ does not decay to zero, but remains at a steady fraction of the original charge:

$$\bar{q}_\infty(\gamma = 1) = \frac{q_0}{1 + \frac{\alpha N_\infty}{Z}}. \tag{A16}$$

This counter-intuitive result, namely persistence of mean charge on a spatially homogeneous, initially charged aerosol, even in perfectly symmetric ion system, is fundamentally a conundrum. This anomaly can only be corrected by introducing, in a fundamental way, space-charge induced drift effects as we show below.

The phenomenon of space charge induced drift does not require spatial gradients in concentrations or charge densities, and hence is consistent with the assumption of spatially homogeneous system. When we introduce space-charge induced drift, Eqs. (A1,A2) get modified as follows.

$$\frac{\partial n^+}{\partial t} = -\mu^+ n^+ \nabla.E + S - \alpha_0 n^+ n^- - n^+ \sum_{q=-\infty}^{\infty} \beta_q^+ Z_q \tag{A17}$$

$$\frac{\partial n^-}{\partial t} = \mu^- n^- \nabla.E + S - \alpha_0 n^+ n^- - n^- \sum_{q=-\infty}^{\infty} \beta_q^- Z_q \tag{A18}$$

$$\frac{\partial Z_q}{\partial t} = -qeBZ_q\nabla.E + n^+ \beta_{q-1}^+ Z_{q-1} + n^- \beta_{q+1}^- Z_{q+1} - (n^+ \beta_q^+ + n^- \beta_q^-)Z_q \tag{A19}$$

Where $B$ is the mechanical mobility of the particle. The Poisson's equation in air for the space-charge induced electric field is

$$\nabla.E = \frac{e}{\epsilon_0}\rho \tag{A20}$$



The time rate of change of the space charge density $\rho$ defined by Eq. (A4 is no longer zero as in Eq.(A8); instead, it assumes the form

$$\frac{d\rho}{dt} = -(\mu^+ n^+ + \mu^- n^- + Z\bar{q^2}eB)\nabla.E \tag{A21}$$

Upon replacing $\nabla.E$ in Eq.(A21) using Eq.(A20), we obtain

$$\frac{d\rho}{dt} = -(\mu^+ n^+ + \mu^- n^- + Z\bar{q^2}eB)\frac{e}{\epsilon_0}\rho \tag{A22}$$

Since the coefficient of the right hand side(rhs) will always be positive non-zero regardless of whether ions are symmetric or otherwise because so long as ion productions exists, ion concentrations will persist. Then Eq. (A22) implies that the space

charge will decay exponentially to zero. i.e.

$$\rho = \rho_0 e^{-(\mu^+ n^+ + \mu^- n^- + Z\bar{q^2}eB)\frac{e}{\epsilon_0}.t} \tag{A23}$$

Hence the final space-charge density $\rho(\to \infty) = \rho_\infty = 0$ regardless of the initial space-charge. This is equivalent to the assumption of quasi-neutrality condition, a key result that is guaranteed only with the inclusion of space-charge effect. With this, from Eq. (A4), we have

$$\bar{q}_\infty = -\frac{1}{Z}(n_\infty^+ - n_\infty^-) \tag{A24}$$

As in Eq. (A12), second relationship for the mean charge $\bar{q}_\infty$, follows from the steady-state charge distribution, including ion asymmetry ($\gamma$):

$$\bar{q}_\infty = \frac{1}{\alpha}[ln\frac{n_\infty^+}{n_\infty^-} + ln\gamma], \tag{A25}$$

where $\alpha = \frac{r_c}{a}$. To solve Eqs.(A24,A25), we introduce quantities representing net ion density ($D_{ions}$) and total ion density

($2N$) as follows:

$$D_{ions} = \frac{n_\infty^+ - n_\infty^-}{n_\infty^+ + n_\infty^-} \tag{A26}$$

and

$$N_\infty = \frac{n_\infty^+ + n_\infty^-}{2} \tag{A27}$$

With Eqs.(A26,A27), the Eqs. (A24,A25) transform to



$$\bar{q}_\infty = -(\frac{2N_\infty}{Z})D_{ions} \qquad (A28)$$

and

$$\bar{q}_\infty = \frac{1}{\alpha}[ln\frac{1+D_{ions}}{1-D_{ions}} + ln\gamma] \approx \frac{1}{\alpha}[2D_{ions} + ln\gamma] \qquad (A29)$$

for weak ion asymmetry( $D_{ions} \ll 1$). Upon eliminating $\phi$, the solution to Eqs. (A28,A29) is

$$\bar{q}_\infty = \frac{ln\gamma}{(\alpha + \frac{Z}{N_\infty})} \qquad (A30)$$

This expression is identical to the $\gamma$-dependent part of the mean charge expression obtained without invoking space-charge effect Eq. (A15). It however, gets rid of the persistence of original charge ($q_0$) . As a result, for perfectly symmetric system of ions ( $\gamma = \frac{\mu^+}{\mu^-} = 1$), unlike Eq. (A15,A16), it follows from Eq. (A30) that $\bar{q}_\infty = 0$, i.e. the aerosol system will eventually undergo complete neutralization regardless of the relative concentrations of particles and ions. The introduction of space-charge effect at a conceptual level guarantees neutralization for an ideal situation of symmetric ions. In a realistic case of ion asymmetry ($\gamma \neq 1$),

a nonzero value of aerosol mean charge $\bar{q}_\infty$ persists, which decreases gradually as $\frac{Z}{N_\infty} \to \infty$. These matters are consistent with the results of Hoppel and Frick (1990).

*Code and data availability.* The model code and simulation data are available upon request.

.

*Author contributions.* YSM conceived the idea and built the initial framework. RP, YSM and KG set up the analytical formulations. KG
developed a numerical version and implemented the model. KG did the calculations and analysed the results with contributions from YSM. KG, RP and YSM wrote the manuscript. All authors approved the final text.

*Competing interests.* The authors declare that there is no conflict of interest.

*Acknowledgements.* The authors gratefully acknowledge the financial support provided by Board of Research in Nuclear Science (BRNS), Department of Atomic Energy (DAE), Government of India to conduct this research under project no. 36(2,4)/15/01/2015-BRNS.



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
