# Peer review of "Impact of space charge on neutralization efficiency of highly charged aerosols: analytical and numerical insights using bipolar ions"

_Aerosol Research, 2024_

## Author Comment (AC1)

**We appreciate the effort of the reviewers and editor in reviewing our manuscript. Our detailed responses are provided below in blue.**

**Editor Comment:**

I regret to inform you that the reviewers of your manuscript have advised against its publication.

The three reviewers report that this work requires drastic improvements to be considered for publication in Aerosol Research. Together with major concerns and many comments/suggestions, the reviewers stated that:

The model and the study are not progressed enough to justify the publication.

We strongly disagree with this comment. Our model represents the first fully 2D-scale approach to exploring the space charge effect on the neutralizer efficiency of aerosol neutralizers. To the best of our knowledge, only a limited number of studies exist on this topic, and these employ simpler models, such as the 1.5D-scale approach by Jidenko et al. (2021, DOI: https://doi.org/10.1080/02786826.2021.1984384). Additionally, no substantial evidence has been provided by the reviewers to indicate that this specific area of study has already been sufficiently explored.

The manuscript requires improvements regarding aerosol flows, particle charging and fluid dynamics.

We agree that the manuscript can be improved in terms of presenting the results related to aerosol flows, particle charging, and fluid dynamics.

The paper shows several critical issues that undermine its validity and discourage its publication

We disagree with the assertion that the critical issues raised by the reviewers undermine the validity of our results. For instance, Reviewer 1's primary concern appears to be a 'misconception of neutralized aerosols.' Specifically, they state: 'In your results, you show that an aerosol, where each particle carries 10 positive charges, is brought to a condition where every single particle in a 2D model is electrically neutralized by exposure to bipolar ions.'

We strongly disagree with this interpretation. We are fully aware of the steady-state charge distribution, often referred to as the Boltzmann (or modified Boltzmann) charge distribution, that governs aerosols under bipolar charging conditions. This concept is explicitly discussed multiple times in the manuscript, including in the main text and the Appendix (e.g., see lines 31, 39, 42, 249, 286, and 326). A similar pattern is observed with other reviewers' comments, which we address in detail below. We

believe these issues do not significantly affect the validity of our results, and we are confident in the robustness of our findings.

The study does not provide significant novelty to the aerosol community.

We also disagree with the comment regarding the lack of significant novelty in our study. Our work introduces a novel perspective on the role of space-charge effects in aerosol neutralization. Specifically, we demonstrate that using a single-particle 'birth and death' model alone predicts the decay of the initial particle charge to zero when $n^+ = n^-$. However, by incorporating the ion balance equation for consistency, we reveal that the net space charge does not decay to zero (i.e., quasi-neutrality is not attained), even when ion-ion and ion-aerosol interactions are accounted for.

Remarkably, our analysis shows that quasi-neutrality is achieved, and the decay of particle charge begins only when space-charge-induced ion drift is considered—something neither ion recombination nor aerosol attachment alone can explain. While the space-charge effect has minimal influence on the neutralization rate for dilute aerosols beyond predictions from single-particle models, it is essential for understanding the existence of neutralization itself.

To the best of our knowledge, such an analysis has not been presented in the literature, marking a significant and novel contribution to the aerosol community.

Therefore, based on these reports, the manuscript cannot be accepted for publication, and it has to be rejected.

**Referee Comment 1:**

general                                                                      comments:

 - The authors present a computational model to simulate the neutralization of a highly charged aerosol, furthermore they present results for four cases, where one different parameter was varied for every case, to show the influence of the selected parameter.

 - In general the model and the study are not progressed enough to justify a

publication, as can be seen from my comments below.

 - Major improvement of readability and language are required, several statements are unclear and some terms are used inaccurately. In general, readability can be improved.

We acknowledge the need to improve the readability and clarity of the manuscript. We will carefully review the text to address unclear statements and ensure accurate usage of terminology throughout. Additionally, we will refine the language to enhance overall readability and coherence.

 - I want to suggest to revise introduction, especially the first part and the part describing neutralization (i.e. charging behavior of neutralizers is standardized in ISO 15900) and the motivation. I mention specific suggestions in the specific comments section below.

The focus of this work is to examine the particle concentration and space-charge effects on the neutralization rate of particles in finite systems. This has not been thoroughly examined in the literature. Additionally, we will incorporate your recommendation to align the description of neutralization with ISO 15900 standards and refine the motivation section to better articulate the context and significance of our study.

 - In order to promote consistent terminology I suggest to use "$N_{i}t$-product" rather than "Nt product"

Nt product was a terminology that was in vogue in the seventies and eighties. We appreciate the suggestion to use "$N_{i}t$-product" for consistent terminology. We will update the manuscript accordingly.

 - It seems that your model does not consider critical influences as, e.g., aerosol transport, ion-ion interactions, ion-particle interactions and diffusional mixing of the gas. So it is not clear to me how you can draw general conclusions from this model.

I think there is a slight misunderstanding. If you look at our equations (1-6), you will recognize the terms which account for transport, ion-ion recombination and ion-particle interactions and diffusive mixing. For instance, in Equation (1), the second term on the right-hand side represents the ion-ion interaction, while the third term corresponds to ion-aerosol (particle) interactions. Similarly, all critical terms relevant to neutralizing an aerosol system have been included, along with an additional space charge term, which is typically not considered in development of the neutralizer or

converter aerosol charge distribution  unknown to known (Ref: Chen et al., 2024, DOI: https://doi.org/10.1080/02786826.2024.2359558).

We will state it explicitly in the text to ensure that it is communicated properly.

Additionally, we acknowledge an editorial error in Equation (6), which will be corrected in the revised manuscript to ensure consistency with Equation (4).

 - The results are very weak for a stand-alone peer-review paper. I'd suggest to move the air-flow profile into supplementary or an appendix, such simulations are very basic. The ion-profile calculation does not seem to represent the reality accurately, as you do not consider several influences like ion-ion interactions and diffusion.

Thank you for your feedback. We agree that the presentation of the results can be further strengthened. Specifically, we would like to highlight the space-dependent neutralization behaviour, dependence on particle concentration and effect of initial ion concentration. We intend to present the results in graphical variations.  For this, we are planning to conduct additional analysis in the revised manuscript to provide a more comprehensive discussion.

However, your last comment above is not correct as, our model does incorporate ion-ion interactions and relevant processes in its calculations. Therefore, the current ion profile accurately reflects the output of the model based on these considerations.

We agree that air flow profiles could be moved to the supplementary materials or an appendix to streamline the main text.

Furthermore the difference between the 1 lpm and 5 lpm case seems rather extreme. Also it is not clear to me why there are already ions at the inlet and how the ion concentration does not increase between the inlet and the outlet, if have implemented a time-dependent production rate in your model. As the results in the section 3.2 are building up on the aforementioned, I am very critical about the significance of this study considering this point alone.

Thank you for your thoughtful comments. The flow rates selected in our study (1 to 5 lpm) are well within the range of those used by commercially available neutralizers. Similar flow rates have also been employed in other studies on aerosol neutralizers, such as the work by Ibarra et al. (2019, DOI: https://doi.org/10.1016/j.jaerosci.2019.105479).

We agree with the reviewer's observation that ions should not be present at the inlet and should instead build up spatially. However we made this assumption to be in

conformity with an earlier study, wherein ions are considered homogeneously distributed throughout the cylinder, with no spatial variation (see Figure 2 in Ibarra et al., 2019).

To address this concern, in the revised manuscript, we will incorporate additional simulations considering different inlet ion concentration varying from zero to maximum profiles to evaluate their impact on the neutralizing efficiency. This will provide a more detailed analysis and strengthen the understanding of neutralization processes.

 - The point I am most critical about is the misconception of neutralized aerosols. In your results you show that an aerosol, where each particle carries 10 positive charges is brought to a condition where every single particle in a 2D-model is electrically neutralized by exposure to bipolar ions. This is not the case in reality, neutralized aerosols have a net-zero charge, single particles are charged, this is why differential mobility analyzers (DMA) operate with a neutralizer - these devices would not work if particles would be uncharged after a neutralizer.

We strongly disagree with this comment. We are fully aware of the existence of the steady-state charge distribution, often referred to as the Boltzmann (or modified Boltzmann) charge distribution, on aerosols due to bipolar charging. This concept is explicitly mentioned multiple times in the manuscript, including in the main text and the Appendix (e.g., see sentences on line 31, line 39, line 42, line 249, line 286, and line 326). The relaxation of particle mean charge to zero is a pointer to the fact that the charge distribution on particles as a whole has relaxed to equilibrium charge distribution. That is why our calculations focus solely on the mean charge of aerosols with an implicit understanding that it ensures attainment of Boltzmann distribution. These are demonstrated in equations A16, A25, and A30. The results and plots presented in the manuscript are derived from this mean charge, not the absolute charge of individual particles. Consequently, this comment does not apply to our work as presented.

To avoid any further misunderstandings, we will ensure that the description of the steady-state charge distribution and the distinction between mean charge and absolute charge is clarified and emphasized in the revised manuscript.

 - To be published the model and the manuscript require improvements regarding aerosol flows, particle charging and fluid dynamics.

Thank you for the valuable feedback and we will improve the manuscript accordingly.

specific                                                                    comments:

17: I would recommend toning down the statement, that mobility size selection is the most precise and reliable one. Although DMAs are the most common devices for size selection, this might also be because of the convenient products available. Please also mention alternative approaches for particle size selection and analysis for the sake of completeness.

We agree with this suggestion and will take care of it.

22: A more convenient way to classify aerosol chargers is to separate them into direct and indirect (diffusion) chargers, e.g. UV and X-ray chargers use very different mechanisms for aerosol charging. I.e. UV charging ist mostly used for direct charging of particles, rather then generate ions.

We agree and the text will be modified accordingly. We will revise the manuscript to incorporate this classification, explicitly highlighting the differences between UV charging (direct) and X-ray charging (indirect).

24: "ionizer" is not a common term in this context, I suggest to use "charger" or "ion source"

We will replace "ionizer" with "charger" or "ion source".

25-26: This statement is either incomplete or inaccurate - please specify of which process the two possibilities are possible, or if you want to keep it general add possible ion-ion interactions.

This sentence was intended to explain how the discharging of charged particles and the charging of neutral particles compete with each other to bring about relaxation to steady-state charge distribution in a bipolar ion environment. We will improve this.

28-29: please specify the challenge for highly charged aerosols, or remove the sentence#

The conventional estimates of neutralization rates via Nt product is a dilute system limit and fails for highly charged aerosols. The initial space charge, the particle concentration will limit the power of the ion source and will significantly alter the rates. We will add this clarification in our updated manuscript.

29-34: This passage is unclear in terminology, please clarify the difference between electrically neutral particle and a neutralized aerosol

(we are not sure where it occurs)

We will clarify that an electrically neutral particle refers to a particle with no net charge, while a neutralized aerosol refers to an aerosol whose charge has been reduced or balanced, typically through ion exchange or other neutralization processes.

35: please complete the list of biplar chargers and please explicitly mention that corona chargers are actually unipolar chargers and the alternating operation is operation mode which allows to release both charge polarities

We have clearly mentioned that bipolar ion sources are based on either radioactive sources or AC corona chargers. We will list soft X-ray bipolar ion sources in addition. We will explain further how an AC corona source acts as a bipolar source.

44: replace "say" by "e.g."

We agree.

45-47: The description using the Nit-product for the description of the charge distribution and the (size dependent) average charge per particle is based on doi:10.1080/02786828808959180 and the "birth-and-death-model" of doi:10.1016/0004-6981(70)90052-1

We will add the references.

50-51: The satement that the mentionen theories demonstrated the size dependent behavior is wrong, please revise wording. Theories tend to describe obeservations of physical processes, so if theories indicate a behavior then because they were developed to do so.

We will modify the sentence accordingly

63 - 69: The connection between the single statements is not clear, please revise the consistency of the paragraph.

We will revise the paragraph to improve clarity and ensure a consistent connection between the statements.

75-79: This general overview is very nice. Please add details about the aerosol (e.g. concentration, particle-size-distribution, material properties) and how the particles are distributed in the tube.

Thank you for your positive feedback. We will add details about the aerosol, including its concentration, particle size distribution, material properties, and how the particles are distributed within the tube, to provide a more comprehensive overview.

85-87: the description of variables is incomplete

We will add the missing variable description.

93-101: I suggest to revise this passage and especially reconsider the statement regarding the "neutralization conundrum" - it rather seems that a model not considering space charge is incomplete. First, the behavior was already described over 30 years ago, as you even indicate in your sources. Second, the case of a absent space charge is physically impossible - a system with a non-zero net-charge will always bring a space charge (the electromagentic force has an infinite range, although it decays quadratically with the distance). Third you even state in Appendix "A": "[...] space-charge-induced drift is the guarantor of complete neutralization [...] and not merely the symmetry of the ions." - this does not sound like a conundrum, more like a understood physical system.

If the word conundrum is not acceptable, we will change it to "elucidation" or equivalent. However, we are certain that our derivation is new and non-trivial. It arises like this.

If you use a single particle "birth & death" model,  you will arrive at decay of initial particle charge to zero for the case $n^+ = n^-$.  However, if one combines ion balance equation for consistency, then one notices that the net space charge does not decay to zero (i.e. quasi-neutrality not attained)  even if ion-ion and ion-aerosol interactions are invoked. As a result, the particle mean charge does not decay to zero. Fascinatingly, quasi-neutrality is attained, and particle charge decay gets initiated only when we invoke space-charge induced drift of ions (not recombination or aerosol attachment).  In quantitative terms, the space-charge effect hardly influences the neutralization rate (beyond that gotten from the single particle model) for dilute aerosols; however, it is a must for understanding the existence of neutralization. This kind of analysis has not been done in the literature and hence our analysis deserves to be published.

101: The importance of the section "Ion and charge aerosol dynamics" is not clear at this point, also it is unclear if you consider space charges in your model from here on or not. Furthermore you did not inculde the explenation of how you are considering aerosol dynamics (or even transport) in your model

We will include details on the implementation of aerosol and charge dynamics within the model, as well as clarify whether space charges are considered.

Figure 1: Please extend the caption in general and describe the symbols in the figure.

We will extend the caption to provide a more detailed explanation and include descriptions of all symbols in the figure.

115-116: Please revise this statement, the simplification to 2D does not require forced flow. Please reconsider the use of the term "convection"

We will revise the statement to clarify that the simplification to 2D does not require forced flow and reconsider the use of the term "convection" to ensure it is accurate and appropriate.

123: Please specify the boundary conditions

We will specify the boundary conditions in detail to ensure clarity and completeness.

123: Why do you use a turbulent model? The flow at conditions in table 1 is laminar

We agree with your observation and will update the model accordingly. A simple parabolic velocity profile, as used in Ibarra et al., 2019, will suffice for this case.

124: I assume you meant "cell" where you wrote "grid"

Yes, we meant to wrote cell, we will correct this accordingly.

134: Why did you choose these flow rates? Why do you expert differences at this conditions?

The flow rate was chosen based on realistic neutralizer aerosol flow rates, as mentioned in Ibarra et al., 2019. Our results show that varying the flow rate significantly impacts neutralizer efficiency when considering space charge effects. Typically, neutralizer design parameters do not account for space charge effects (see, e.g., Chen et al., 2024), which can lead to inaccuracies in predicting the final size distribution.

142 - 145: This statement is not clear. I interpret that you meant most likely, that what is shown in fig. 3 are the four flows after the neutralizer, which have been exposed to aerosols at 10E4 #/ccm and a ion production rate of 10E6 #/(ccm s), but for the shown flow the aerosols are not present any more and no ions are produced, also diffusional mixing does not seem to happen. I do not understand why this graphs are of importance, also I am not completely sure if I understood it correctly. Also I am wondering why you only show positive ions and no effects of ion-ion interaction are visible, while you were talking about bipolar charging and various interactions when explaining your model.

We agree with the reviewer's point regarding this figure. This figure shows the steady-state ion concentration profile after the charged aerosol passes through the cylinder, which does not reflect the time-dependent ion and charged aerosol dynamics of our model. We will remove this figure in the revised manuscript and provide a more in-depth analysis of the ion concentration profiles for both positive and negative ions under various scenarios.

151-222: I only give general comment on this technical corrections:

- There is a major misconception of neutralized aerosols. It is not necessary for an neutralized aerosol, that every single particle is electrically neutral. See ISO 15900.

We quite understand that neutralization implies that the aerosol system as a whole will attain a steady-state Boltzmann (or modified Boltzmann) charge distribution (this has been stated various places in the text), the mean charge of which will decay to zero. The relaxation dynamics of mean-charge is a good indicator that an equilibrium distribution has been attained and in view of this we focus on mean charge as a means of input to design neutralizers.

**Referee Comment 2:**

The paper reports a numerical simulation of neutralisation of charged aerosol using bipolar ions, a subject that clearly fall within the scope of this journal and which tackles a subject of interest for its audience.

The research question behind the paper derives from the need to support current understandings of the aerosol neutralisers for high concentration aerosols, and the authors aims to present their model as a tool to explore the basic discharging mechanisms of neutralisers as well as the the way how spatial heterogeneity of ions affect the neutralisation rate.

The paper title is accurate, the related works seems reported in an adequate way, the paper structure is adequate and the style is appropriate, although it may benefit a revision to become more fluent.

Despite the interesting topic, in my opinion the paper shows several critical issues that undermine its validity and discourage its publication at this stage. In particular:

1. The methodologies proposed here lack for adequate details of the numerical modelling and are based on assumptions that cannot be accounted for easily. Among that:

Thank you for your comment. We will provide additional details on the numerical modelling methods used and address the assumptions made in the methodology to improve clarity and transparency.

i) a planar 2D simulation as the one used here can be used to approximate the behaviour of a channel, but is not appropriate to describe a cylinder, for which axial-symmetric schemes should be used;

Thank you for the comment. We have already used an axial-symmetric scheme in our simulation, which assumes that the system's properties are invariant around the central axis and reduces the problem to radial and axial coordinates. This approach is appropriate for modelling cylindrical geometries, and we will clarify this in the revised manuscript.

ii) the boundary conditions in the domain are not clear: what is the potential at the walls? What about the inlet and outlet sections?;

We will specify the boundary conditions in detail to ensure clarity and completeness.

iii) what kind of numerical modelling is used to solve the fluid dynamic field, the aerosol motion and the electric field equations?

We have used a semi-implicit scheme to solve the aerosol motion and electric field equations, as described in our previous work (Ghosh et al., 2017, 2019, and 2020). We will include additional details on the numerical modelling in the revised manuscript.

iv) what kind of turbulence model is used and why this is needed?

We have used a k-epsilon based turbulence model; however, since the flow is considered laminar in this work, we will simplify the model by using a parabolic flow rate profile, similar to that used in Ibarra et al., 2019.

v) how the ions interacts among them and with the walls?

In our ion dynamics scheme (equations 1 and 2), we account for ion-ion interactions, ion-aerosol interactions, ion diffusion, ion mobility influenced by space charge effects, and convective processes due to the inflow of air.  Walls are treated as absorbers of ions upon contact.

The paper presents a severe lack of clarity in the methodology section and this limits the readibility and the credibility of the results. The methodology has to be carefully revised and the results updated consequently.

Thank you for your valuable feedback. We will carefully revise the methodology to ensure a clearer and more detailed explanation of the approach used.

2. The simulated conditions seem not representative of realistic neutralisers or the assumptions behind them are not clear:

The simulation conditions used in our study fall within the working parameter range typically assumed for this type of neutralizer, as supported by other works in the field (Ibarra et al., 2019; Chen et al., 2024). We will revise the manuscript to include more detailed explanations regarding the selection of the simulation conditions to address this issue.

i) with the simulated diameter and length of the cylindric neutraliser, how it is possible to maintain a constant and spatially uniform production of the ions? Can they be achieved for whatever pressure, temperature, humidity and composition of the gases? Should the ion source positioning respect to the aerosol flow be realistically neglected? Why this scheme can be considered a general one? If the simulation is considering "idealised" neutralization conditions, how its results can be extended to the many possible applications cited in the Conclusion?

Usually neutralizers use Kr-85 type of pure beta emitters which uniformly irradiate the air volume and hence act as uniform ion production sources. The source positioning is done to expose maximum air volume to ensure uniform ion-production rate and has little impact on aerosol flow. We are following the assumptions provided by the neutralizers, which are standardized in ISO 15900. We will include details about the humidity, pressure, and temperature conditions in our revised manuscript to provide greater clarity.

As for why this scheme can be considered general, the methodology we employ is designed to be applicable across a broad spectrum of conditions, given its focus on fundamental ion dynamics and aerosol-neutralization principles. This generalization is supported by previous studies (e.g., Ibarra et al., 2019; ISO 15900), which have shown that similar approaches can be adapted to various neutralizer configurations.

While the simulation does focus on idealized neutralization conditions, we believe that the results can still be extended to many practical applications. The idealized conditions help to isolate key processes and provide a foundational understanding, which can then be adapted or modified for real-world applications with specific adjustments to parameters.

We will update the manuscript to clarify these points and provide a more detailed justification for the general applicability of the simulation results.

ii) the aerosol size is not specified, so that the assumed number of charge of 10e charge cannot be compared with its Pauthenier's limit: are 10e a high or a small fractions of the ions a particle can have? Besides, the aerosol size influences its dynamics in the neutraliser due to the different Brownian diffusivity and the relative effects on thermal and electrophoretic motions. Are the particles considered massless?

In our study, we used 1-micron particles, which is reasonable for carrying 10 elementary charges. The Pauthenier limit applies to field charging, which differs from the diffusion charging mechanism used here.

While we have included Brownian diffusivity in the model (Eq. 6, first term), its effect is relatively small compared to the convection induced by the inflow rate, making it negligible for this study. We account for electrophoretic motions through space charge effects, and thermal motion is ignored due to our assumption of constant temperature. The particles are not massless, and we will update the manuscript accordingly for clarity.

The authors must carefully revise the methodology section and provide more details and explanations on these points.

3. The effect of aerosol concentration is critical, but is treated here in a rather simplified way while, according to the ambitions of the work, this should be one its main topic. What is the contribution of this model in understanding for what conditions the "classical Nt product concept does not hold" [lines 203-204]? I suggest extending the discussion on this topic, after revision of the aforementioned points 1 and 2.

We acknowledge that the effect of aerosol concentration is indeed crucial, and its treatment in the current manuscript is oversimplified. To address this, we plan to expand the discussion on the limitations of the "classical Nt product" concept, particularly focusing on conditions where it may not apply. We will revise the manuscript to provide a more detailed analysis of aerosol concentration and its role in neutralizer dynamics, which will be one of the central topics in the updated version.

4. Most of the other findings of the model are intuitive and not new: for example it is clear from the basic physics that higher flow rates means lower neutralization time, higher ions concentration fasten the neutralization.

The intent of our work is not to re-demonstrate the known effects of flow rate or ionization rate on neutralization efficiency. Instead, our focus is on highlighting the significance of the space charge effect under specific conditions.

For example, as shown in Figure 4 of the manuscript, at flow rates of 3 and 5 LPM, the space charge effect prevents ions from neutralizing aerosols in the upper and lower boundary regions. In cases of limited ions or short residence time, minor anomalies in ion distribution caused by the space charge effect can significantly impact neutralizer efficiency.

To address this, we will include a comparison of results with and without the space charge effect in the revised manuscript to clarify our findings.

5. The conclusions of the paper seem not sustained by the methodologies and the result, especially the last paragraph that extend the validity of the paper findings without having any substantial basis.

We understand that the conclusions drawn in the paper, especially in the final paragraph, need to be more rigorously supported by the methodologies and results presented. We will revise the conclusions to ensure they are directly linked to the data and analyses performed and will clarify any extrapolation made in the final section to make it consistent with the study's findings. The revised manuscript will present a more robust connection between the results and conclusions.

**Referee Comment 3**

General Comment:

The paper investigates the effect of space charge on the neutralization efficiency of charged aerosols, exploring variables such as flow rate, ion production rate, aerosol concentration, and neutralizer geometry. Although the research question is relevant, the findings lack originality, as many results—such as the impact of aerosol concentration and ion production rate—are known to aerosol scientists. The study does not provide significant novelty to the aerosol community.

For publication, I recommend a major revision of the paper. Significant improvements are needed in the methodology, data presentation, and scientific discussion to increase the clarity and depth of the findings. In addition, the conclusions need to be rewritten to better reflect the results and provide more meaningful contributions to the field.

1) The introduction is not logically structured and contains repetitive points. A clearer structure is recommended, including introduction of charged aerosol, discussion of aerosol neutralization (concept and need), summary of recent research, identification of research gaps, and statement of purpose and novelty of the study.

We thank the reviewer for pointing out the need for a clearer and more structured introduction. In the revised manuscript, we will make the necessary changes to reorganize the introduction.

2) Methodology: The methodology is inadequately described, with important variables such as electric field, ionic interactions, and environmental conditions not fully explored. It is also unclear to me the 2D model, how the bipolar ionizer is positioned.

We appreciate the reviewer's observation and acknowledge the need for a more detailed description of the methodology. As suggested by this and other reviewers, we will revise this section in the manuscript to include a comprehensive explanation of the electric field, ionic interactions, and environmental conditions used in the model.

For clarity, the 2D model simulates charged aerosol and ion dynamics in both the horizontal (X-axis) and vertical (Y-axis) planes using an axisymmetric scheme. For simplicity, we do not model the bipolar ionizer as a single point source within the cylinder. Instead, we assume that ions are generated homogeneously throughout the cylinder. These assumptions and their implications will be clearly detailed in the revised manuscript.

3)                        Results                        and                        discussion:

Reliance on only one type of figure/graph in the results weakens the presentation of the data; more graphs showing more novelty data would increase clarity (perhaps neutralization efficiency over time). Also, for the figures presented in the paper, standardization of color scales is needed.

We appreciate the reviewer's suggestion to improve the presentation of the results. In the revised manuscript, we will diversify the types of graphs to better highlight the novelty and clarity of our findings. For instance, we will include additional plots, such as neutralization efficiency over time, to provide a more comprehensive understanding of the data. Furthermore, we will ensure that the color scales across all figures are standardized for consistency and better readability. These changes will enhance the clarity and impact of the results section.

Sections 3.2.1 and 3.2.2 require more explanation, such as why above 3 lpm the neutralizer effectiveness is lower than expected, what is actually expected? What are the references?

We appreciate the reviewer's observation. We agree that clarification is needed for why neutralizer efficiency appears lower than expected above 3 LPM in our findings. Specifically, our results demonstrate that when the space charge effect is considered under these conditions, the neutralizer's efficiency decreases compared to cases where this effect is neglected. This highlights an aspect often overlooked during the design of such neutralizers. We will revise the manuscript to explain this point in detail and include appropriate references to support our claims.

For section 3.2.2, what is the threshold at which high ion production negatively affects neutralization? The simultaneous effects of space charge distribution and velocity need further analysis, particularly regarding their influence on particle neutralization rates.

We appreciate the reviewer's interest in the threshold at which ion production rates negatively affect neutralization. It may be noted that this is the first work alerting hitherto unknown consequences due to space charge effects. Hence it may not be possible to provide practical criteria for designing an optimum neutralizer for dense aerosols. Separate studies would be required for this. For the present, we would like to hightlight the existence of these effects. the influence of factors This threshold depends on various factors, as discussed in our work, including aerosol concentration and the charge level of the aerosol, assuming other neutralizer parameters remain fixed.

Regarding the simultaneous effects of space charge distribution and velocity, we agree that further analysis is needed. In the revised manuscript, we will include a more detailed discussion of their influence on particle neutralization rates, highlighting the interplay between these factors.

In section 3.2.4, the discussion of geometry and neutralizing capacity is unclear, as these results are correlated with other factors analyzed in previous sections. Should we call it a geometry effect?

Thank you for pointing this out. While geometry plays a significant role in neutralizing capacity, we agree that its effects are closely correlated with other factors such as ion production rate, aerosol concentration, and flow dynamics, as analyzed in previous sections. We will clarify in the revised manuscript that the term 'geometry effect' refers to the combined influence of these factors as mediated by the geometric configuration.

4) In addition, the final paragraph of the conclusion does not align with the results presented.

The conclusions are weak and lack depth, not bringing much scientific contribution to the aerosol community.

We appreciate the feedback and will revise the conclusion to better align with the results presented, ensuring it highlights the key scientific contributions of our work to the aerosol community. We will also strengthen the discussion to provide a more in-depth and impactful summary of our findings.

Although the study addresses an interesting topic, it does not offer substantial or new insights. Significant improvements are needed for publication.

However, this work is the first to present a 2D model that highlights the impact of space charge on neutralizer efficiency, supporting similar but simpler (1.5D) modeling studies, such as Jidenko et al. (2021, DOI: https://doi.org/10.1080/02786826.2021.1984384). Furthermore, our work provides substantial theoretical proof (from equations A1 to A30) showing that neglecting space charge in the fundamental charge dynamic equations, as used in neutralizer standards like ISO 15900, can lead to a non-zero mean charge on aerosols.